# Molecular Mechanism Discovery of Acacetin Against Cancers: Insights from Network Pharmacology and Molecular Docking

**DOI:** 10.3390/ijms26199433

**Published:** 2025-09-26

**Authors:** Jung Yoon Jang, Donghwan Kim, Eunok Im, Na Kyeong Lee, Nam Deuk Kim

**Affiliations:** 1Department of Pharmacy, College of Pharmacy, Research Institute for Drug Development, Pusan National University, Busan 46241, Republic of Korea; jungyoon486@pusan.ac.kr (J.Y.J.); eoim@pusan.ac.kr (E.I.); 2Functional Food Materials Research Group, Korea Food Research Institute, Wanju-gun 55365, Jeollabuk-do, Republic of Korea; kimd@kfri.re.kr; 3Department of Pharmacy, Eson Geriatric Hospital, Ulsan 44955, Republic of Korea

**Keywords:** acacetin, cancer, network pharmacology, molecular docking, EGFR, STAT3, AKT

## Abstract

Acacetin, a naturally occurring flavonoid, has attracted increasing attention due to its broad anticancer potential. In vitro and in vivo studies using diverse tumor models have demonstrated that acacetin modulates oncogenic signaling, suppresses angiogenesis, and induces apoptosis and other regulated cell death pathways. With the rising demand for multi-target therapeutics, network pharmacology and molecular docking have emerged as powerful tools to unravel the complex molecular mechanisms of phytochemicals. Unlike previous reviews that have mainly focused on single pathways or limited cancer contexts, this review emphasizes novelty by integrating network pharmacology with molecular docking and explicitly linking these computational predictions to experimental validation, thereby identifying epidermal growth factor receptor (EGFR), signal transducer and activator of transcription 3 (STAT3), and the serine/threonine kinase AKT (also known as protein kinase B (PKB) as central experimentally supported targets. This integrative framework maps acacetin’s multi-target anticancer mechanisms and clarifies its translational opportunities for future therapeutic development.

## 1. Introduction

Cancer is one of the most formidable global public health challenges, consistently ranking as the second leading cause of death worldwide and affecting millions of individuals and families across diverse populations [1,2,3,4,5]. Cancer is characterized by uncontrolled cellular proliferation, invasion of surrounding tissues, and the potential for distant metastasis, all of which together contribute to its high morbidity and mortality [6].

Despite significant advances, traditional cancer therapies, including surgery, chemotherapy, radiation therapy, and hormonal treatments, continue to grapple with intrinsic limitations such as incomplete tumor eradication due to heterogeneity, systemic toxicity, development of drug resistance, and adverse effects on patients’ quality of life [1,7]. Although these modalities remain the mainstay of oncology, therapeutic barriers underscore the urgent need for novel and selective interventions [8]. Moreover, innovative strategies such as targeted therapies, immunotherapies, gene therapy, and nanomedicine have emerged, offering improved specificity and promise for precision medicine. However, obstacles such as antigenic variation, manufacturing complexity, limited tissue penetration, and high costs are still encountered [9]. In this context, bioactive natural compounds, particularly flavonoids, are attracting research interest due to their multi-target actions, favorable safety profiles, and ability to regulate complex oncogenic signaling networks [10]. However, the degree of scientific attention within this class has varied considerably. A PubMed search conducted in August 2025 retrieved 7966 publications for luteolin and 7570 for apigenin, two flavonoids that are structurally related to acacetin, but only 715 for acacetin, indicating that acacetin has been investigated at roughly one-tenth the level of these well-established flavonoids. Despite this gap, emerging evidence has revealed novel anticancer activities of acacetin, particularly through molecular docking studies that predict multiple oncogenic targets. These findings underscore its potential for translational research and justify a dedicated systematic review.

Consistent with this need, reviews on other flavonoids such as isorhamnetin and prunin have highlighted their ability to modulate major oncogenic pathways, including phosphoinositide 3-kinase (PI3K)/serine/threonine kinase (AKT, also known as protein kinase B, PKB), Janus kinase (JAK)/signal transducer and activator of transcription 3 (STAT3), and mitogen-activated protein kinase (MAPK)/extracellular signal-regulated kinase (ERK) [11,12]. These examples demonstrate how diverse flavonoids exert anticancer activity through overlapping mechanisms, thereby helping to contextualize acacetin among other flavonoid-based anticancer agents.

Acacetin, a naturally occurring flavonoid that belongs to the flavone subclass present in various plant species, has demonstrated diverse pharmacological actions, including anticancer, anti-inflammatory, antioxidant, and hepatoprotective effects through modulation of key intracellular pathways such as MAPK/c-Jun N-terminal kinase (JNK)/ERK, nuclear factor kappa-light-chain-enhancer of activated B cells (NF-κB), nuclear factor erythroid 2-related factor 2 (Nrf2), PI3K/AKT/mechanistic target of rapamycin (mTOR), and cyclooxygenase (COX)-2 [13]. Nonetheless, a detailed understanding of the molecular mechanisms by which acacetin exerts its anticancer activity remains limited owing to the inherently networked and redundant signaling architecture of cancer cells [14].

In recent years, network pharmacology, defined as a systems-level approach that integrates bioinformatics, cheminformatics, and systems biology to map interactions between drugs and multiple molecular targets, has emerged as a powerful system-level framework for predicting interactions between bioactive compounds and disease-associated targets, facilitating the identification of hub proteins and enriched signaling pathways through integrated protein–protein interaction (PPI) networks and pathway analysis [15]. Molecular docking offers a computational method for estimating binding affinities and interactions between compounds and target proteins, providing structural insights to support experimental validation [16,17,18,19,20]. These computational strategies have been successfully applied to phytochemicals such as curcumin in osteosarcoma and herbal extracts in prostate and cervical cancers, highlighting their value in uncovering multi-target anticancer mechanisms [21,22,23].

Therefore, in this review, we aimed to synthesize the pharmacological evidence of acacetin’s anticancer activities in vitro and in vivo and complement this with a combined network pharmacology and molecular docking approach to identify key targets and binding interactions. Such efforts may illuminate the potential of acacetin as a multi-target therapeutic agent across diverse cancer types and may inform future strategies for translational and combinatory treatment development.

In preparing this review, we systematically searched the literature using the keyword “acacetin” in the PubMed and Google Scholar databases. The search covered publications available until August 2025 and was restricted to full-text articles written in English. Emphasis is placed on preclinical investigations, particularly those providing mechanistic insights, reporting in vitro and in vivo pharmacological activities, and presenting experimental evidence supporting the predicted molecular targets.

## 2. Overview of Acacetin

### 2.1. Chemical Structure and Sources

Acacetin (5,7-dihydroxy-4′-methoxyflavone) is an *O*-methylated flavone (C_16_H_12_O_5_) belonging to the flavonoid family (Figure 1). Its A-ring bears hydroxyl groups at C-5 and C-7, whereas the B-ring carries a methoxy group at C-4′, and these are features that shape its physicochemical behavior and target engagement in biological systems [13]. In nature, acacetin acts as an aglycone and glycoside in multiple edible or medicinal taxa. It has been isolated from black locust (*Robinia pseudoacacia*), bee propolis, *Dracocephalum moldavica*, *Turnera diffusa*, and *Betula pendula*, as evidenced by chemical profiling in peer-reviewed studies. These sources include widely used foods or ethnomedicinal plants from which acacetin and its conjugates have been chemically characterized [24,25,26].

### 2.2. Pharmacological Activities of Acacetin

Acacetin exerts diverse pharmacological effects. In oncology, it modulates key oncogenic pathways, including PI3K/AKT, JAK/ STAT3, NF-κB, and MAPK, thereby inducing apoptosis, cell cycle arrest, and inhibition of metastasis [27,28,29,30]. Its anti-inflammatory action has been demonstrated in RAW 264.7 macrophages, where acacetin suppresses lipopolysaccharide (LPS)-induced expression of inducible nitric oxide synthase (iNOS) and cyclooxygenase-2 (COX-2) by inhibiting the NF-κB pathway through reduced inhibitor of nuclear factor kappa-B alpha (IκBα) phosphorylation and blockade of NF-κB nuclear translocation [31]. In cardiovascular research, acacetin selectively inhibits atrial potassium currents, including ultra-rapid delayed rectifier K^+^ current and transient outward, prolonging atrial action potential duration and preventing atrial fibrillation in canine models without affecting ventricular repolarization [32,33]. Furthermore, in apolipoprotein E-deficient mice, acacetin attenuated atherosclerosis by activating the Nrf2 signaling pathway and upregulating the antioxidant enzyme methionine sulfoxide reductase A [34].

Beyond these effects, acacetin exhibits antiviral activity against multiple viruses, including herpes simplex virus type 1, dengue virus type 2, influenza virus, and human immunodeficiency virus, underscoring its broad-spectrum potential [35,36,37,38]. It displays antimicrobial efficacy, particularly against methicillin-resistant *Staphylococcus aureus* and various Gram-positive and Gram-negative bacteria, partly through synergy with antibiotics and inhibition of virulence factors such as sortase A [39,40,41]. Moreover, acacetin exerts anti-obesity effects by suppressing adipogenesis, enhancing lipolysis, and activating AMP-activated protein kinase (AMPK)/sirtuin 1 (SIRT1) signaling, consistent with reduced body weight observed in high-fat diet–induced obese mice [42]. These multifaceted activities highlight acacetin’s potential as a therapeutic candidate for a wide range of chronic diseases.

Among these diverse effects, its role in oncology is particularly notable. While structurally related flavonoids such as apigenin [43] and luteolin [44] also regulate major oncogenic pathways, acacetin demonstrates distinctive advantages by directly targeting EGFR, STAT3, and AKT and by uniquely influencing necroptosis, PD-L1 expression, and angiogenesis [14,45,46,47,48]. This highlights its differentiated pharmacological profile compared with other closely related flavones.

## 3. Pharmacological Evidence of Acacetin’s Anticancer Activity

Several mechanistic and pharmacological studies have evaluated the anticancer activity of acacetin in different tumor types. As outlined in Table 1, these findings indicate that acacetin exerts multi-faceted effects both in vitro and in vivo, including apoptosis induction, proliferation inhibition, cell-cycle regulation, suppression of metastasis, and inhibition of angiogenesis.

### 3.1. In Vitro Effects: Apoptosis, Proliferation, and Cell Cycle Arrest

Acacetin exerts potent proapoptotic activity in diverse cancer cell lines by simultaneously activating the intrinsic (mitochondrial) and extrinsic (death receptor) apoptotic pathways. The intrinsic pathway involves disruption of mitochondrial homeostasis with cytochrome *c* release and activation of initiator caspase-9 followed by effector caspase-3/-7, accompanied by downregulation of anti-apoptotic proteins such as B-cell lymphoma 2 (Bcl-2) and B-cell lymphoma-extra large (Bcl-xL) and, in certain contexts, upregulation of pro-apoptotic mediators including Bcl-2-associated X protein (Bax) and Bcl-2 homologous antagonist/killer (Bak), thereby shifting the balance toward apoptosis [43,49,55,56,57]. The extrinsic pathway is promoted by the upregulation of the first apoptosis signal receptor (Fas, also known as CD95 or APO-1) and Fas ligand (FasL) signaling and the activation of caspase-8, which also engages the mitochondrial pathway via BH3-interacting domain death agonist (Bid) cleavage [55,64].

In breast cancer models, acacetin not only induces classical apoptosis but also triggers receptor-interacting protein kinase 1 (RIP1)-dependent necroptosis through sustained ERK1/2 phosphorylation, suggesting its potential to bypass apoptosis resistance mechanisms [50]. Necroptosis induction is not exclusive to acacetin; for instance, apigenin has been reported to induce necroptosis in mesothelioma and pancreatic cancer, and resveratrol has also been shown to induce necroptosis in prostate cancer [7,43]. These findings indicate that several flavonoids can activate necroptosis depending on tumor context, while acacetin’s activity in breast cancer underscores a unique mechanistic contribution that distinguishes it from other flavonoids.

In colorectal carcinoma, suppression of the Wingless-related integration site (Wnt)/β-catenin/cellular myelocytomatosis oncogene (c-Myc) axis and nuclear β-catenin accumulation is coupled with apoptosis-inducing factor (AIF) translocation from mitochondria to the nucleus, leading to caspase-independent cell death [53].

Acacetin disrupts cell cycle progression in several cancer types, inducing G1 arrest in hepatocellular carcinoma via p53 and p21 upregulation [57,58] and S-phase arrest in colorectal carcinoma through the downregulation of cyclin A and cyclin-dependent kinase 2 (CDK2) [53]. These cell cycle regulatory effects are often accompanied by the inhibition of key pro-survival signaling pathways, including PI3K/AKT, MAPK/ERK, and signal transducer and activator of transcription 3 (STAT3), which are commonly dysregulated in malignant cells [27,45,50,54,65].

### 3.2. Anti-Metastatic and Anti-Angiogenic Activities

Metastasis suppression by acacetin has been demonstrated most clearly in gastric cancer, where the inhibition of epithelial–mesenchymal transition (EMT) is mediated through the downregulation of PI3K/AKT/Snail signaling. This results in the restoration of the epithelial marker E-cadherin and the reduction in mesenchymal markers such as N-cadherin, along with decreased expression and activity of matrix metalloproteinase (MMP)-2 and MMP-9, thereby reversing the EMT phenotype [54].

In non-small cell lung cancer (NSCLC) models, acacetin also attenuates tumor cell invasion by downregulating MMP-2 and urokinase-type plasminogen activator (u-PA) through inactivation of JNK and reduced NF-κB/activator protein-1 (AP-1) DNA-binding activity, ultimately suppressing extracellular matrix degradation [29].

Regarding angiogenesis, acacetin decreases vascular endothelial growth factor (VEGF) and hypoxia-inducible factor-1α (HIF-1α) by inhibiting the AKT/HIF-1α pathway, and this translates into significant suppression of ovarian cancer cell-induced angiogenesis and tumor growth in vivo (as demonstrated in animal models), thereby linking its molecular effects to anti-angiogenic efficacy in tumors [48]. Although comprehensive evaluations are still limited, current evidence suggests that acacetin preferentially suppresses angiogenesis within the tumor microenvironment, where AKT/HIF-1α signaling is aberrantly activated, while exerting relatively minor effects on normal vasculature [66,67]. Further in vivo studies directly comparing tumor-associated and physiological angiogenesis will be necessary to confirm the selectivity of this effect.

### 3.3. In Vivo Efficacy and Safety Profiles

Multiple animal models and human tumor xenografts have been used to confirm the anticancer efficacy of acacetin in vitro [45,47,54,63]. In gastric cancer xenograft models, acacetin treatment significantly reduced tumor volume and weight, consistent with the suppression of the phosphoinositide 3-kinase/protein kinase B/Snail (PI3K/AKT/Snail) signaling pathway [54]. In a separate gastric cancer xenograft study, acacetin administration was associated with a lower Ki-67 proliferation index and increased levels of apoptotic markers, including cleaved caspase-3 and poly(ADP-ribose) polymerase (PARP), in tumor tissues, in parallel with the inhibition of STAT3 signaling. In lung cancer xenograft models, acacetin treatment led to a marked reduction in metastatic nodules in the lungs accompanied by decreased programmed death-ligand 1 (PD-L1) expression, suggesting both antiproliferative and immunomodulatory effects [47]. From an immuno-oncological perspective, PD-L1 suppression by acacetin may also enhance responsiveness to immune checkpoint inhibitors (anti-PD-1/PD-L1 therapies), thereby providing potential synergistic benefits in lung cancer treatment [68,69]. In prostate cancer xenografts, tumor regression is associated with the strong inhibition of STAT3 Tyr705 phosphorylation, achieved through the direct binding of acacetin to STAT3, which contributes to reduced tumor cell proliferation and increased apoptosis in tumor tissues [45]. In a skin cancer xenograft model, acacetin directly targets the p110 catalytic subunit of PI3K, resulting in the inhibition of AKT phosphorylation and significant suppression of tumor growth [63]. Collectively, these in vivo findings underscore the ability of acacetin to inhibit tumor progression across diverse cancer types by modulating multiple oncogenic pathways. Although detailed toxicological evaluations remain limited, no severe adverse effects or significant body weight loss has been reported in the available studies, suggesting a favorable preliminary safety profile [25,45,47,54,63]. Nevertheless, most available studies have relied on conventional xenograft models, which do not fully capture the complexity of human tumors; to enhance translational relevance, patient-derived xenografts (PDX) and organoid systems should be incorporated to more robustly evaluate the anticancer efficacy and safety of acacetin [70,71].

## 4. Network Pharmacology-Based Target Prediction of Acacetin

The predicted and validated targets of acacetin identified in previous network pharmacology and molecular docking studies are summarized in Table 2. This table lists each target protein, associated cancer types, implicated pathways, type of evidence (computational prediction, docking simulation, or experimental validation), functional outcomes, and corresponding references. Notably, several targets such as the epidermal growth factor receptor (EGFR), STAT3, and AKT1 appeared as high-ranking hub genes in network analyses and as experimentally confirmed molecular targets, underscoring the concordance between in silico prediction and empirical data. For clarity, docking scores are reported as binding free energies (kcal/mol); affinities below −6.0 kcal/mol are generally considered favorable for potential pharmacological relevance, whereas values closer to −4.0 kcal/mol indicate weak binding and should be interpreted with caution [72].

### Methodological Pipeline and Identified Targets

Previous network-pharmacology studies investigating acacetin have reported its potential multi-target spectrum in oncology through integrated computational workflows that combine ligand-based target prediction [78,79,80], disease–gene mapping [81,82,83,84], PPI analysis [85], enrichment analysis [86], and, in some cases, molecular docking [81,87]. As summarized in Table 2, these studies identified targets across multiple functional classes, including EGFR [14,46,73], AKT [46,74,77], proto-oncogene tyrosine-protein kinase Src (SRC) [46,77], transcription factors (STAT3 [45,77]), and molecular chaperones or inflammatory mediators such as (heat shock protein 90 alpha family class A member 1(HSP90AA1) [46,77] and tumor necrosis factor (TNF) [46].

In the reported workflows, the compound–target prediction stage is generally conducted using ligand-based virtual screening platforms such as SwissTargetPrediction [78], Search Tool for Interacting Chemicals (STITCH) [79], or the Similarity Ensemble Approach (SEA) [80], that generates candidate protein interactions based on chemical similarity metrics and known bioactivity profiles. Molecular docking simulations were performed using AutoDock Vina [81] to evaluate the binding affinities between acacetin and its predicted targets. To ensure relevance to oncology, predicted protein targets were cross-referenced with cancer-associated genes obtained from databases, including GeneCards (human gene compendium) [82], Online Mendelian Inheritance in Man (OMIM) [83], and the disease–gene association network database (DisGeNET) [84] (e.g., EGFR, AKT1 [46]).

The intersection sets of predicted and disease-related genes were subsequently analyzed through PPI network construction using the Search Tool for the Retrieval of Interacting Genes/Proteins (STRING) database [85], followed by network visualization and topological analysis in Cytoscape [86].

The resulting PPI networks enabled the visualization of interactions among acacetin-associated targets and the identification of nodes with higher connectivity that were considered more likely to play central roles in the pharmacological activity of the compound. Several of these computationally prioritized targets such as EGFR [14,46,73], STAT3 [45,77], and AKT [46,74,77] are also supported by experimental studies as direct or indirect mediators of the anticancer effects of acacetin (see Section 5), thereby indicating concordance between network-based predictions and empirical evidence.

## 5. Molecular Docking and Experimental Validation of Key Targets

As outlined in Section 4 and summarized in Table 2, network pharmacology analyses highlighted several acacetin-associated targets of potential oncological relevance. In this section, we focus on the targets for which molecular docking simulations and experimental studies provide complementary evidence.

### 5.1. Overview of Key Network-Derived Targets

Acacetin was further examined against several network pharmacology–highlighted targets, most notably EGFR [14,46,73], STAT3 [45,77], and AKT [46,74,77], by using molecular docking simulations and experimental validation (Table 2). These proteins consistently emerged as central nodes in the predicted protein–protein interaction networks and were mechanistically linked to critical oncogenic signaling pathways, including PI3K/AKT, JAK/STAT, and MAPK/ERK [14,45,46,73,74,77].

### 5.2. EGFR

EGFR molecular docking using the crystal structure of the EGFR kinase domain predicted a stable binding conformation for acacetin within the kinase active site that was characterized by a favorable docking score. Target engagement was further supported by experimental validation using drug affinity responsive target stability (DARTS) and cellular thermal shift assays (CETSA), both of which demonstrated a direct interaction between acacetin and EGFR in gastric cancer cells [88,89]. Consistent with these in silico and biophysical findings, in vitro assays demonstrated that acacetin treatment reduced EGFR phosphorylation and suppressed activation of downstream STAT3 and ERK pathways, leading to decreased proliferation. In vivo, acacetin administration in gastric cancer xenografts significantly inhibited tumor growth without marked systemic toxicity [14]. In addition to these gastric cancer findings, docking studies in colorectal cancer models predicted a strong binding affinity of −8.3 kcal/mol [46], while bladder cancer models exhibited a binding affinity of −4.0 kcal/mol [73]. According to docking benchmarks, affinities below −6.0 kcal/mol are often considered indicative of favorable interactions, whereas values around −4.0 kcal/mol represent relatively weak binding with limited pharmacological relevance [72]. This distinction highlights the importance of interpreting docking scores in the context of biological significance and experimental validation. Taken together, these results further support the potential of acacetin as a broadly relevant EGFR-targeting compound across multiple tumor types.

### 5.3. STAT3

For STAT3, docking simulations demonstrated strong binding of acacetin to the SH2 domain of STAT3, stabilized by three hydrogen bonds and a cation–π interaction. Biophysical assays, including DARTS, CETSA, and pull-down experiments, confirmed a direct interaction between acacetin and STAT3 in prostate cancer DU145 cells. In vitro, acacetin treatment decreased p-STAT3 (Tyr705) levels, downregulated downstream STAT3-regulated anti-apoptotic and cell cycle–related proteins such as cyclin D1, Bcl-2, Bcl-xL, Mcl-1, and Survivin, and increased pro-apoptotic Bax expression. These effects were accompanied by PARP and caspase-3 cleavage, Annexin V positivity, and elevated reactive oxygen species (ROS) generation. In vivo administration of acacetin to DU145 xenograft models significantly suppressed tumor growth and reduced STAT3 Tyr705 phosphorylation. Collectively, these findings indicate that acacetin functions as a STAT3 inhibitor and is a potential drug candidate for targeting STAT3 in cancer therapy [45]. Furthermore, network pharmacology and molecular docking in gastric cancer models predicted a strong binding affinity of acacetin for STAT3 (−9.0 kcal/mol), reinforcing computational evidence for STAT3 as a potential target across multiple tumor types [77].

### 5.4. AKT

For AKT1, docking simulations predicted that acacetin occupies the ATP-binding pocket of the kinase domain, forming a hydrogen bond with Thr211 and exhibiting a strong binding affinity (−9.2 kcal/mol) that is indicative of potential competitive inhibition. Experimental validation in colorectal cancer HT-29 cells confirmed these predictions, indicating that acacetin treatment reduced p-AKT and p-PI3K levels, leading to the downregulation of Survivin and decreased cell migration and proliferation. Acacetin increases p53 expression and induces caspase-3 cleavage, resulting in enhanced apoptosis. These molecular effects are consistent with the inhibition of the PI3K/AKT/p53 signaling axis, supporting AKT1 as a direct target of acacetin [46]. Additionally, in lung cancer models, acacetin inhibited HSP90AB1, a molecular chaperone that stabilizes AKT, leading to reduced p-AKT and p-mTOR levels, suppression of EMT markers, decreased migration and invasion, and restoration of E-cadherin expression. These effects were reversed by the HSP90 agonist terazosin [74]. Additionally, gastric cancer–based network pharmacology and molecular docking analyses reported a notable binding affinity of acacetin for AKT1 (−8.3 kcal/mol) [77], providing complementary computational support for its potential role as an AKT1-targeting compound across diverse malignancies.

### 5.5. Additional Network-Derived Targets

In addition to EGFR, STAT3, and AKT1, acacetin was associated with several other targets, and this was supported primarily by docking predictions, with a subset corroborated by experimental validation. In colorectal cancer–focused analyses, SRC exhibited a docking affinity of −6.9 kcal/mol, the docking affinity of HSP90AA1 was −7.1 kcal/mol, and that of TNF was −6.4 kcal/mol [46]. In bladder cancer models, IL-6 and MYC exhibited relatively weaker predicted affinities of −4.12 kcal/mol and −4.37 kcal/mol, respectively [73]. Among the targets with functional corroboration, prostaglandin-endoperoxide synthase 2 (PTGS2) in nasopharyngeal carcinoma demonstrated < −7 kcal/mol docking with a key Tyr385 contact and facilitated in vitro suppression of proliferation and migration [76]. In skin cancer models, acacetin directly targeted PI3K (p110), as validated by pull-down/kinase assays and xenografts, and docked to the ATP-binding site (H-bonds: Val828, Glu826, and Asp911; hydrophobic: Trp760, Ile777, and Tyr813) [63]. In breast cancer systems, PI3Kγ inhibition by acacetin was supported by kinase and Western blot assays, while docking analyses indicated binding at the ATP site through hydrogen bonds (Ser806, Ala885, and Val882) and hydrophobic interactions (Lys833 and Asp964) [75]. Additionally, a gastric cancer–based network pharmacology study reported “high” docking affinities (mostly < −7.3 kcal/mol) for a broader panel that included STAT3, AKT1, MAPK1, HSP90AA1, HRAS, SRC, PIK3CA, PIK3R1, FYN, and RHOA [77]. Although many of these associations remain supported primarily by computational predictions, they provide testable hypotheses where systematic investigation may reveal whether acacetin exerts pleiotropic anticancer effects via the concurrent modulation of multiple oncogenic signaling pathways. Nonetheless, in silico approaches such as docking and network pharmacology are inherently limited by the quality of structural data and algorithmic assumptions, and their results do not always translate into biologically relevant interactions without experimental validation [16,72]. 

## 6. Challenges and Future Prospects

Despite a growing body of preclinical evidence supporting the anticancer potential of acacetin, several challenges remain to be addressed before its effective clinical translation.

### 6.1. Bioavailability and Pharmacokinetics

Acacetin exhibits very low aqueous solubility and poor oral absorption, with instability across physiological pH levels and gastrointestinal fluids, resulting in limited intestinal uptake. Consequently, its oral bioavailability in rats is extremely low (~2%), while intravenous studies have demonstrated rapid plasma clearance and a short half-life (~1.5 h) [90,91]. These findings highlight the urgent need for the development of advanced formulation strategies.

### 6.2. Nanoparticle-Based Delivery and Structural Optimization

The pharmacokinetic limitations of acacetin have spurred the exploration of nanotechnology-based delivery approaches, including polymeric nanoparticles, liposomes, and solid lipid nanoparticles that can enhance bioavailability, prolong circulation time, and increase tumor selectivity via enhanced permeability and retention effect [90,92,93,94]. Surface modification with tumor-targeting ligands (e.g., folate, RGD [Arg-Glu-ASP] peptides, and antibodies) has demonstrated reduced off-target toxicity in preclinical studies, suggesting the potential of acacetin formulations [95,96,97]. Parallel to delivery strategies, structural optimization through medicinal chemistry, such as prodrug derivatization, may improve the stability and absorption, distribution, metabolism, and excretion (ADME) properties [98,99,100,101]. Collectively, nanoparticle-mediated delivery and structural modification represent synergistic strategies to overcome the pharmacokinetic barriers of acacetin and enable its development into clinically viable formulations [102,103].

### 6.3. Synthetic Derivatives and Structural Analogs of Acacetin

Synthetic derivatives and structural analogs of acacetin have been developed to overcome its pharmacokinetic limitations and enhance its anticancer efficacy. Aminoalkyl and Mannich base derivatives of acacetin-7-*O*-methyl ether demonstrated enzyme inhibitory activity, supporting the feasibility of structural diversification [104]. Simple glycosylated analogs such as linarin and linarin acetate exhibited much weaker activity than that of acacetin, indicating that certain glycosylation patterns can reduce anticancer potency [62]. In contrast, semisynthetic 7-*O*-*β*-*D* glycosides of acacetin prepared from naringin exhibited measurable cytotoxicity across multiple cancer cell lines (HL-60, SMMC-7721, A549, MCF-7, and SW480), suggesting that specific glycosylation patterns may preserve or even restore biological activity [105]. Methylated flavone derivatives exhibit improved hepatic metabolic stability and intestinal absorption compared to their unmethylated counterparts, supporting methylation as a viable strategy for orally available analogs [98,106]. In parallel, biotechnological methods enable scalable production of methylated flavonoids with favorable pharmacological profiles [100]. Collectively, these findings highlight the ability of methylation to consistently improve pharmacological stability, whereas the effects of glycosylation are context-dependent. In some cases, glycosylation reduces activity, whereas in others, it can yield analogs with favorable pharmacological profiles. These structural modifications complement the formulation strategies for enhancing the translational potential of acacetin [62,98,100,105].

### 6.4. Synergistic Drug Combinations

The most direct evidence of synergy comes from non-small-cell lung carcinoma, where acacetin enhances the therapeutic efficacy of doxorubicin by reducing clonogenic survival, inducing G2/M arrest, and suppressing multidrug resistance protein 1 (MDR1)-mediated efflux, thereby increasing the intracellular retention of doxorubicin [107]. Beyond this, although acacetin’s inhibition of EGFR [14,46,73], STAT3 [45,77], and PI3K/AKT [46,74,75,77] signaling, as well as its suppression of PD-L1 expression [47] and VEGF/HIF-1α–mediated angiogenesis [48], are mechanistically compatible with combination strategies, formal synergy studies remain lacking. In this context, combinations with clinically relevant chemotherapeutics such as cisplatin or paclitaxel appear theoretically plausible and warrant systematic evaluation in preclinical models. Thus, a systematic evaluation of well-designed preclinical models is required to establish the role of acacetin as an adjuvant in combination therapies, both in vitro and in vivo.

### 6.5. Need for Clinical Translation

Despite accumulating preclinical evidence demonstrating the anticancer efficacy of acacetin in vitro and in various animal models, no clinical trials have evaluated its safety, tolerability, or therapeutic effectiveness in humans, and this represents a critical translational void. Most research remains confined to cell culture and xenograft models that report pro-apoptotic, anti-proliferative, and anti-metastatic effects; however, these effects remain clinically unverified [25,108]. Human pharmacokinetic and toxicity data are lacking, precluding the determination of dosing regimens, dose-limiting toxicities, or metabolic pathways [25,109]. Therefore, early-phase safety studies are essential for clinical advancements [110].

## 7. Conclusions

Acacetin has exhibited promising preclinical anticancer activity, as supported by extensive in vitro and in vivo studies, and has been systematically characterized using an integrated approach combining network pharmacology and molecular docking. Mechanistic investigations have demonstrated that acacetin suppresses tumor progression by inhibiting oncogenic signaling and angiogenesis while inducing apoptosis, cell cycle arrest, anti-metastatic activity, and other regulated cell death modes. These effects are mediated primarily through EGFR, STAT3, and AKT, as well as additional critical pathways. Molecular docking and experimental validation further confirmed the ability of acacetin to directly target key oncogenic proteins such as EGFR, STAT3, and AKT, underscoring its multi-target potential (Figure 2).

Despite this progress, significant challenges remain in clinical translation. Poor aqueous solubility, rapid systemic clearance, and metabolic instability hinder effective bioavailability, highlighting the need for innovative delivery systems, such as nanoparticles, and rational structural optimization. Additionally, synthetic derivatives and structural analogs of acacetin, particularly glycosylated and methylated forms, have been investigated to improve metabolic stability and absorption, and in certain cases cytotoxic potency, underscoring structural modification as a complementary strategy for future drug development. Moreover, while preliminary studies suggest synergy with standard chemotherapeutics such as doxorubicin, systematic evaluation of acacetin in well-designed preclinical combination models and early-phase clinical trials is still required.

Taken together, the current evidence suggests that acacetin holds promise as a multi-target anticancer agent with potential applications in personalized medicine and combination therapies. At the same time, tumor heterogeneity may critically influence the clinical translation of such multi-target activities, underscoring the need to evaluate acacetin’s efficacy across diverse genetic and molecular tumor contexts. From a safety standpoint, although preclinical studies have not reported severe toxicity, potential off-target effects on normal cells and tissues cannot be fully excluded. Accordingly, comprehensive toxicology and long-term safety evaluations should accompany efficacy studies. Future research should prioritize translational studies integrating advanced formulation strategies, structural modifications of synthetic derivatives, rational drug combinations, and biomarker-driven patient stratification to establish acacetin as a clinically viable anticancer therapeutic agent.

## Figures and Tables

**Figure 1 ijms-26-09433-f001:**
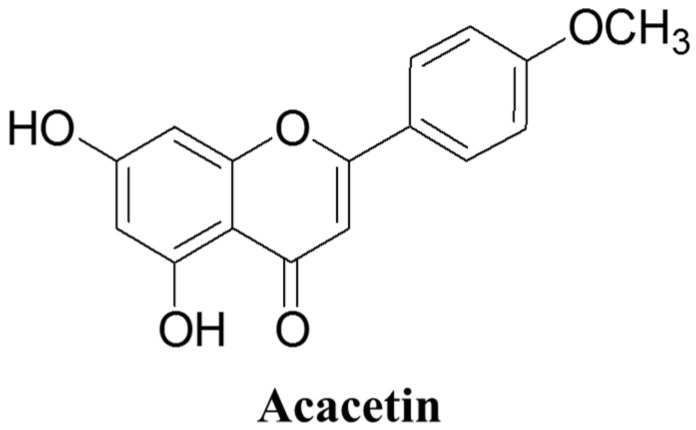
Molecular structure of acacetin.

**Figure 2 ijms-26-09433-f002:**
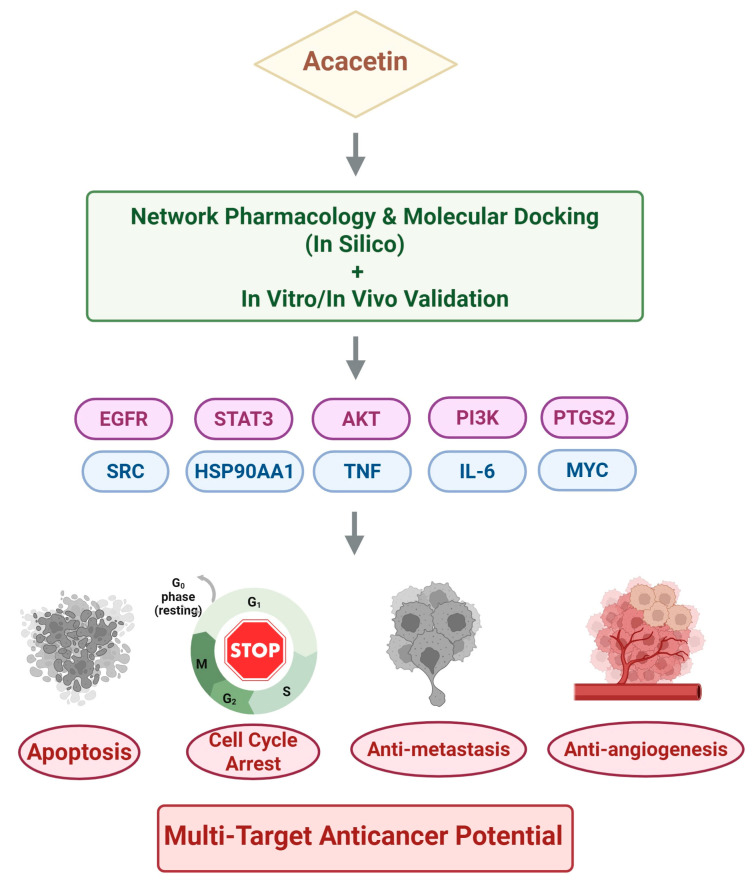
Schematic overview of the multi-target anticancer mechanisms of acacetin based on network pharmacology, molecular docking, and experimental validation. Acacetin exerts anticancer activity through an integrated approach combining network pharmacology and molecular docking (in silico) with in vitro and in vivo validation. These analyses identified multiple oncogenic protein targets, including EGFR, STAT3, AKT, PI3K, and PTGS2 (validated by docking, in vitro, and in vivo studies), as well as SRC, HSP90AA1, TNF, IL-6, and MYC (supported by docking only), which collectively regulate critical signaling pathways. Functionally, acacetin induces apoptosis, promotes cell cycle arrest, and exerts anti-metastatic and anti-angiogenic effects, thereby underscoring its potential as a multi-target anticancer therapeutic. Figure created with BioRender.com. AKT, serine/threonine kinase (also known as protein kinase B, PKB); EGFR, epidermal growth factor receptor; HSP90AA1, heat shock protein 90 alpha family class A member 1; IL-6, interleukin-6; MYC, myelocytomatosis oncogene; PI3K, phosphoinositide 3-kinase; PTGS2, prostaglandin-endoperoxide synthase 2 (COX-2); SRC, proto-oncogene tyrosine-protein kinase Src; STAT3, signal transducer and activator of transcription 3; TNF, tumor necrosis factor.

**Table 1 ijms-26-09433-t001:** Summary of acacetin’s anticancer effects by cancer type.

Interventions Cancer Type	Cell Line(s)/Model	Up-Regulation	Down-Regulation	Mechanisms	Refs.
Breast	T-47D, MDA-MB-231, MCF-7, MDA-MB-468,	DNA fragmentation, sub-G1 cells, Cyt *c* (cytosolic) AIF (cytosolic), p-SAPK/JNK1/2, p-c-Jun, JNK1/2, ROS, cleaved caspase-7,-8 and -9, cleaved PARP, p-ERK, Chk1, Bax, Mitochondrial superoxide generation, RIP1, RIP3 p21, p27	Bcl-2, MMP, Cyt *c* (mitochondrial), AIF (mitochondrial) CDK1, CDK2, Cdc25C, Cyclin B1, Cyclin E, AKT, FADD, p-AKT, AKT, p38, ERK	Apoptosis, Necroptosis, G2/M arrest, ERK activation Migration and EMT inhibition	[49,50,51]
Colorectal	HT-29, HCT 116, SW480	ROS, MMP (mitochondrial), AIF	β-catenin, c-Myc,	Apoptosis, S and G2/M arrest	[52,53]
Esophageal	TE-1, TE-10	Bax	Ki-67, MMP-2 and -9, Bcl-2, p-JAK2, p-STAT3	Apoptosis, JAK2/STAT3 inhibition, Migration suppression	[28]
Gastric	MKN45, MKN45 xenograft, MGC803, AGS	E-cadherin, Bax, cleaved PARP, cleaved caspase-3, DNA fragmentation, Sub-G1 cells, ROS, Cyt *c* (cytosolic) caspase-3,-8, and -9 activity, Fas, FasL, cleaved Bid, p53	N-cadherin, MMP-2 and -9, Snail, p-PI3K, p-AKT, p-EGFR, Bcl-xL, p-STAT3, p-ERK, p-EGFR, PCNA, DFF-45, MMP	Apoptosis, EMT suppression, Anti-metastasis, PI3K/AKT/Snail inhibition	[14,54,55]
Liver	HepG2, HepG2/RARγ xenografts	DNA fragmentation, cleaved PARP, p21, p53, FasL, mFasL, sFasL, caspase-8 activity, Bax, cleaved caspase-3,	p-STAT3, p-JAK1,p-JAK2, p-AKT, p-Src, Cyclin D1, Bcl-2, Bcl-xL, Mcl-1, Survivin, Mcl-2, VEGF, p-IκBα (cytosolic), p-p65 (cytosolic), pro-casapase-3, RARγ, p-GSK-3β, Ki67	Apoptosis, STAT3 inhibition, Invasion inhibition, Angiogenesis inhibition, G1 arrest, Apoptosis via non-genomic RARγ-AKT-p53 pathway	[56,57,58]
Lung	A549, H460, A549 xenografts	Bak, p53, miR-34a, IκBα (cytosolic), G1 phase, p21, Fas, FasL	Cyclin B1, Cyclin D, Bcl-2, PD-L1, MMP-2 and -9, u-PA, p-p38α, p-MKK3, 6, p-MLK3, NF-κB, NF-κB p50 (nuclear), NF-κB p65 (nuclear) AP-1, c-Fos, c-Jun, p-IκBα (cytosolic)	Proliferation, invasion, and migration inhibition, G2/M arrest, Apoptosis	[29,30,47,59]
Ovarian	SKOV3, OVCAR-3, A2780, A2780 (CAM assay in vivo model)	AKT	PCNA, MMP-2 and -9, IL-6 and -8, RAGE, p-PI3K, p-AKT, VEGF, HIF-1α	Cell proliferation and invasion inhibition, Apoptosis, RAGE-PI3K/AKT inhibition, Angiogenesis inhibition, AKT/HIF-1α inhibition	[48,60]
Osteosarcoma	HOS	Bax, cleaved PARP, cleaved caspase-3, -8, and -9, Cyt *c*, ROS, p-c-Jun, c-Jun, p-JNK,	Colony-formation, Bcl-2, Survivin, MMP	Apoptosis, ROS/JNK activation	[61]
Prostate	DU145, DU145 xenograft, LNCaP	p53, IκBα, Bax, cleaved PARP, sub-G1 cells, ROS, p-JAK2, p-p38, p21	p-AKT, p-GSK-3β, p-NF-κB p65, p-IκBα, Bcl-2, XIAP, COX-2, p-STAT3 (Y705), Cyclin D1, Bcl-2, Bcl-xL, Mcl-1, Survivin, STAT3 activity, CDK2, CDK4, CDK6, Cdc25C, Cdc2, Cyclin B1	Apoptosis, STAT3 inhibition	[27,45,62]
Skin (Melanoma)	SK-MEL-28, SK-MEL-28 xenograft	G1 arrest	p-AKT (Thr308), p-AKT (Ser473), p-GSK3β, Cyclin D1, Tumor volume	PI3K p110 binding, PI3K/AKT/p70S6K inhibition	[63]

AIF, apoptosis-inducing factor; AP-1, activator protein 1; AKT, serine/threonine kinase (also known as protein kinase B, PKB); Bax, Bcl-2-associated X protein; Bak, Bcl-2 homologous antagonist/killer; Bcl-2, B-cell lymphoma 2; Bcl-xL, B-cell lymphoma-extra large; CAM assay, chorioallantoic membrane assay; CDK, cyclin-dependent kinase; COX-2, cyclooxygenase-2; Caspase, cysteine-aspartic protease; Chk1, checkpoint kinase 1; Cyt *c*, cytochrome *c*; DFF-45, DNA fragmentation factor 45; E-cadherin, epithelial cadherin; EGFR, epidermal growth factor receptor; EMT, epithelial–mesenchymal transition; ERK, extracellular signal-regulated kinase; FADD, Fas-associated death domain protein; Fas, first apoptosis signal receptor (CD95/APO-1); FasL, Fas ligand; GSK-3β, glycogen synthase kinase-3 beta; HIF-1α, hypoxia-inducible factor 1-alpha; IL-6, interleukin-6; IκBα, inhibitor of nuclear factor kappa-B alpha; JAK2, Janus kinase 2; JNK, c-Jun N-terminal kinase; MKK, mitogen-activated protein kinase kinase; MMP, mitochondrial membrane potential; MMP-2, matrix metalloproteinase-2; Mcl-1, myeloid cell leukemia-1; NF-κB, nuclear factor kappa-light-chain-enhancer of activated B cells; p-, phosphorylated; PARP, Poly(ADP-ribose) polymerase; PCNA, proliferating cell nuclear antigen; PD-L1, programmed death-ligand 1; PI3K, phosphoinositide 3-kinase; RAGE, receptor for advanced glycation end-products; RARγ, retinoic acid receptor gamma; RIP, receptor-interacting protein kinase; ROS, reactive oxygen species; SAPK, stress-activated protein kinase; STAT3, signal transducer and activator of transcription 3; VEGF, vascular endothelial growth factor; XIAP, X-linked inhibitor of apoptosis protein; u-PA, urokinase-type plasminogen activator.

**Table 2 ijms-26-09433-t002:** Confirmed and predicted molecular targets of acacetin based on docking and network pharmacology.

Target	Cancer Type(s)	Function	Docking/Prediction/Validation Method	Validation Level	Functional Outcome	Refs.
EGFR	Gastric	EGFR-mediated signaling (STAT3, ERK)	Target prediction (SwissTargetPrediction, GeneCards, RNA-seq); network analysis (STRING, Cytoscape); docking (Schrödinger Maestro); validation (DARTS, CETSA, WB, TUNEL, colony formation, xenograft)	Confirmed (docking + in vitro + in vivo)	↓: p-EGFR, p-STAT3, p-ERK (transient), Ki-67, PCNA, Bcl-xL, Colony formation, Tumor growth	[14]
					↑: Bax, cleaved caspase-3, cleaved PARP, Apoptosis,	
					Strong EGFR binding (docking)	
	Colorectal	-	Target prediction (SwissTargetPrediction, TCMSP, GeneCards, OMIM, DisGeNET); network analysis (STRING, Cytoscape); docking (AutoDock Vina)	Docking only	Binding affinity −8.3 kcal/mol	[46]
	Bladder	EGFR-mediated signaling	Target prediction (TCMSP, GeneCards, OMIM); network analysis (STRING, Cytoscape); docking (AutoDock Vina)	Docking only	Binding affinity −4.0 kcal/mol	[73]
STAT3	Prostate	STAT3-mediated signaling (proliferation, survival)	Docking (Glide, Schrödinger Maestro); binding validation (DARTS, CETSA, pull-down); functional assays (WB, Annexin V/PI, xenograft)	Confirmed (docking + in vitro + in vivo)	↓: p-STAT3 (Tyr705), Cyclin D1, Bcl-2, Bcl-xL, Mcl-1, Survivin, Tumor volume,	[45]
					↑: Bax, cleaved PARP, cleaved caspase-3, Annexin V, p-JAK2, p-p38, ROS, Apoptosis	
					Strong STAT3 SH2 binding (3 H-bonds + cation–π; Glide)	
AKT1	Colorectal	PI3K/AKT/p53 signaling	Target prediction (SwissTargetPrediction, TCMSP, GeneCards, OMIM, DisGeNET); network analysis (STRING, Cytoscape); docking (AutoDock Vina); validation (WB, CCK-8, Annexin V-FITC, scratch assay)	Confirmed (docking + multiple in vitro assays)	↓: p-AKT, p-PI3K, Survivin, Migration, Proliferation	[46]
					↑: p53, cleaved caspase-3, Apoptosis,	
					Strong AKT1 binding (−9.2 kcal/mol; 1 H-bond, Thr211)	
HSP90AB1 → AKT	Lung	Chaperone-mediated PI3K/AKT/mTOR activation	Network pharmacology; docking (AutoDock Tools); validation (WB, EMT, invasion assay ± terazosin)	Confirmed (docking + in vitro + in vivo)	↓: p-AKT, p-mTOR, Migration, Invasion, EMT markers	[74]
					↑: E-Cadherin; effect reversed by HSP90 agonist terazosin	
PI3Kγ	Breast	PI3K/AKT/mTOR/p70S6K/ULK signaling	Docking (SYBYL2.0); PI3Kγ kinase assay; validation (WB [PI3Kγ, p-AKT, p-mTOR, p-p70S6K, p-ULK1], Apoptosis assay, G2/M arrest, LC3 puncta)	Confirmed (docking + in vitro)	↓: PI3Kγ, p-AKT, p-mTOR, p-p70S6K, p-ULK1	[75]
					↑: Apoptosis, Autophagy, G2/M arrest	
					PI3Kγ ATP-site binding (H-bonds: Ser806, Ala885, Val882; hydrophobic: Lys833, Asp964)	
SRC	Colorectal	Predicted PI3K/AKT signaling; proliferation/migration in CRC	Target prediction (SwissTargetPrediction, TCMSP, GeneCards, OMIM, DisGeNET); network analysis (STRING, Cytoscape); docking (AutoDock Vina)	Docking only	Binding affinity −6.9 kcal/mol	[46]
HSP90AA1	Colorectal	Predicted chaperone-mediated PI3K/AKT; cell survival in CRC	Target prediction (SwissTargetPrediction, TCMSP, GeneCards, OMIM, DisGeNET); network analysis (STRING, Cytoscape); docking (AutoDock Vina)	Docking only	Binding affinity −7.1 kcal/mol	[46]
TNF	Colorectal	Predicted inflammatory/apoptotic signaling; CRC progression	Target prediction (SwissTargetPrediction, TCMSP, GeneCards, OMIM, DisGeNET); network analysis (STRING, Cytoscape); docking (AutoDock Vina)	Docking only	Binding affinity −6.4 kcal/mol	[46]
IL-6	Bladder	Inflammatory cytokine signaling	Target prediction (TCMSP, GeneCards, OMIM); network analysis (STRING, Cytoscape); docking (AutoDock Vina)	Docking only	Binding affinity −4.12 kcal/mol	[73]
MYC	Bladder	Cell proliferation and transcription	Target prediction (TCMSP, GeneCards, OMIM); network analysis (STRING, Cytoscape); docking (AutoDock Vina)	Docking only	Binding affinity −4.37 kcal/mol	[73]
PTGS2	Nasopharyngeal carcinoma	Inflammation, Tumor proliferation and migration	Target prediction (TCMSP, SwissTargetPrediction, SEA, GeneCards, OMIM); network analysis (STRING, Cytoscape); docking (AutoDock Vina), MD simulation; validation (CCK-8, colony formation, migration, WB [PTGS2])	Confirmed (docking + in vitro)	↓: PTGS2 (docking: <−7 kcal/mol; Tyr385 contact), Proliferation, Migration	[76]
PI3K (p110 subunit)	Skin	PI3K-AKT-p70S6K signaling	Binding assays (pull-down, kinase); cell transformation (in vitro); docking (Glide/induced-fit, Maestro); xenograft	Confirmed (docking + in vitro + in vivo)	↓: PI3K activity, p-AKT, p-p70S6K, Tumor growth, Colony formation,	[63]
					↑: G1 arrest; docking to PI3K-p110 ATP-binding site (H-bonds: Val828, Glu826, Asp911; hydrophobic: Trp760, Ile777, Tyr813)	
STAT3, AKT1, MAPK1, HSP90AA1, HRAS, SRC, PIK3CA, PIK3R1, FYN, RHOA,	Gastric	PI3K-AKT, MAPK, JAK/STAT3, and cytoskeleton regulation pathways	HPLC-Q-TOF–MS/MS identification; network pharmacology (STRING, Cytoscape); docking (AutoDock Vina)	Docking only	High binding affinity (mostly < −7.3 kcal/mol)	[77]

AKT1, RAC-alpha serine/threonine-protein kinase (protein kinase B); Bax, Bcl-2-associated X protein; Bcl-2, B-cell lymphoma 2; Bcl-xL, B-cell lymphoma-extra large; CCK-8, cell counting kit-8; CETSA, cellular thermal shift assay; CRC, colorectal cancer; Cytoscape, Cytoscape network visualization software; DARTS, drug affinity responsive target stability; EGFR, epidermal growth factor receptor; EMT, epithelial–mesenchymal transition; HSP90AA1, heat shock protein 90 alpha family class A member 1; JAK2, Janus kinase 2; MAPK, mitogen-activated protein kinase; Mcl-1, myeloid cell leukemia-1; MD, molecular dynamics; OMIM, Online Mendelian Inheritance in Man; PARP, poly(ADP-ribose) polymerase; PCNA, proliferating cell nuclear antigen; PI3K, phosphoinositide 3-kinase; PTGS2, prostaglandin-endoperoxide synthase 2 (cyclooxygenase-2); Pull-down assay, biotinylated compound-based target validation assay; RNA-seq, RNA sequencing; ROS, reactive oxygen species; SEA, similarity ensemble approach; Schrödinger Maestro, Schrödinger molecular modeling suite; SRC, proto-oncogene, non-receptor tyrosine kinase Src; STAT3, signal transducer and activator of transcription 3; STRING, Search Tool for the Retrieval of Interacting Genes/Proteins; SwissTargetPrediction, SwissTargetPrediction in silico target prediction tool; TCMSP, Traditional Chinese Medicine Systems Pharmacology database; TNF, tumor necrosis factor; TUNEL, terminal deoxynucleotidyl transferase dUTP nick end labeling; WB, Western blot. ↑, increase; ↓, decrease; →, mechanistic upstream regulatory relationship between proteins.

## Data Availability

Data presented in this study are available upon request from the corresponding author.

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
