# Peer review of "Molecular Mechanism Discovery of Acacetin Against Cancers: Insights from Network Pharmacology and Molecular Docking"

_ijms, 2025, doi:10.3390/ijms26199433_

Round 1
Reviewer 1 Report
Comments and Suggestions for Authors
This review summarizes the anticancer mechanisms of acacetin by integrating network pharmacology, molecular docking, and experimental validation. The manuscript highlights EGFR, STAT3, and AKT as primary targets and presents preclinical evidence of apoptosis induction, anti-metastasis, and anti-angiogenic effects. It further discusses pharmacokinetic challenges, nanoparticle delivery, and structural analogs. The scope is valuable; however, the manuscript requires major revisions in terms of novelty framing, literature depth, critical analysis of computational predictions, and figure presentation.
Comments for the authors
Comment 1. Emphasize in the abstract how this review advances beyond existing reviews on flavonoids or acacetin, make novelty explicit.
Comment 2. Expand discussion in the introduction by positioning acacetin among other flavonoid-based anticancer agents and directly compare mechanistic overlaps.
Comment 2. Integrate more recent literature in the introduction, especially on flavonoids modulating oncogenic signaling. (see:10.3390/ijms26062678; 10.3390/ijms26157381).
Comment 3. Add discussion on whether docking affinities (e.g., -4.0 kcal/mol in bladder cancer EGFR prediction) are biologically meaningful or too weak to imply pharmacological relevance.
Comment 4. Clarify whether the necroptosis pathway reported in breast cancer is unique to acacetin or shared with other flavonoids, expand on translational significance.
Comment 5. Provide more detail on the role of PD-L1 modulation in lung cancer and its potential in immuno-oncology.
Comment 6. Discuss whether inhibition of angiogenesis via AKT/HIF-1α is selective to tumor endothelium or also affects normal vasculature.
Comment 7. Comment on the limitations of xenograft models and propose patient-derived xenografts or organoid systems for better translational insight.
Comment 8. Add a table or sub-section comparing acacetin with other flavonoids (e.g., apigenin, luteolin, prunin, isorhamnetin) in terms of network pharmacology targets and binding affinities.
Comment 9. Improve the conclusion section by outlining a translational roadmap.
Comment 10. Edit the manuscript for grammar and scientific tone. Correct tense inconsistencies, remove redundancy, and refine word choice for precision. Professional language editing is required.
Author Response
- Point 1: Emphasize in the abstract how this review advances beyond existing reviews on flavonoids or acacetin, make novelty explicit.
- Response 1: We appreciate this valuable comment. In the revised abstract, we explicitly highlight the novelty of our review by emphasizing the integration of network pharmacology, molecular docking, and experimental validation that converge on EGFR, STAT3, and AKT as central targets. We also clarify in the Introduction that, unlike apigenin and luteolin, acacetin remains comparatively underexplored despite emerging evidence of distinctive anticancer activities, thereby justifying a dedicated systematic review.
- Point 2: Expand discussion in the introduction by positioning acacetin among other flavonoid-based anticancer agents and directly compare mechanistic overlaps.
Integrate more recent literature in the introduction, especially on flavonoids modulating oncogenic signaling. (see:10.3390/ijms26062678; 10.3390/ijms26157381).
- Response 2: Thank you for the helpful suggestion. We expanded the Introduction to position acacetin among related flavonoids and to compare mechanistic overlaps with structurally similar agents (e.g. apigenin, luteolin). We also incorporated the suggested recent reviews on isorhamnetin and prunin (3390/ijms26062678; 10.3390/ijms26157381), emphasizing how flavonoids modulate PI3K/AKT, JAK/STAT3, and MAPK/ERK pathways. These additions improve continuity and clarify acacetin’s place within the flavonoid family.
- Point 3: Add discussion on whether docking affinities (e.g., -4.0 kcal/mol in bladder cancer EGFR prediction) are biologically meaningful or too weak to imply pharmacological relevance.
- Response 3: We appreciate this insightful comment. We now discuss the biological interpretation of docking affinities, noting that values around around -4.0 kcal/mol (e.g. bladder cancer EGFR prediction) generally indicate limited pharmacological relevance, whereas affinities below approximately -6.0 kcal/mol are more consistent with meaningful interactions – particularly when supported by experiments. We revised the EGFR section accordingly and added a benchmarking reference (4. Network Pharmacology-based Target Prediction of Acacetin, 5.2. EGFR) to contextualize docking scores.
- Point 4: Clarify whether the necroptosis pathway reported in breast cancer is unique to acacetin or shared with other flavonoids, expand on translational significance.
- Response 4: We thank the reviewer for this valuable comment. In the revised manuscript, we have clarified that the induction of necroptosis is not unique to acacetin
(3.1. In Vitro Effects: Apoptosis, Proliferation, and Cell Cycle Arrest). For example, our previous review on apigenin reported necroptosis induction in mesothelioma and pancreatic cancer, and our review on resveratrol also described evidence of necroptosis induction. These observations indicate that necroptosis can be triggered not only by acacetin but also by certain other natural compounds, depending on the tumor context. At the same time, we emphasized that in acacetin-treated breast cancer models, necroptosis is distinctly associated with specific signaling features, such as sustained ERK1/2 activation and RIP1 activation, which highlight acacetin’s unique mechanistic profile. Furthermore, as noted in Section 3.1, we discussed the translational significance of necroptosis induction in tumors that evade apoptosis, underscoring its potential as an alternative therapeutic strategy in preclinical settings.
- Point 5: Provide more detail on the role of PD-L1 modulation in lung cancer and its potential in immuno-oncology.
- Response 5: We appreciate this suggestion. We expanded Section 3.3 to elaborate on PD-L1 modulation in lung cancer and its immuno-oncology implications, noting the potential for synergy with immune checkpoint blockade (anti-PD-1/PD-L1). This addition strengthens the translational implications of PD-L1 modulation in lung cancer.
- Point 6: Discuss whether inhibition of angiogenesis via AKT/HIF-1α is selective to tumor endothelium or also affects normal vasculature.
- Response 6: We thank the reviewer for raising this important point. We clarified that acacetin appears to preferentially inhibit angiogenesis in tumor contexts where AKT/HIF-1α signaling is dysregulated, while normal vasculature may be less affected. However, as direct comparative studies are lacking, future in vivo experiments will be essential. To support this statement, we have added references highlighting the differences between tumor and normal angiogenes (3.2 Anti-Metastatic and Anti-Angiogenic Activities).
- Point 7: Comment on the limitations of xenograft models and propose patient-derived xenografts or organoid systems for better translational insight.
- Response 7: We thank the reviewer for this thoughtful comment. In the revised manuscript, we acknowledged that most current studies investigating the anticancer effects of acacetin have relied on conventional xenograft models. While these models provide valuable in vivo data, they do not fully capture the complexity of human tumor biology, including stromal interactions and immune responses. To address this limitation, we highlighted the need for future preclinical studies employing patient-derived xenografts (PDX) and organoid systems, which can better recapitulate tumor heterogeneity and provide improved translational relevance (3.3. In Vivo Efficacy and Safety Profiles).
- Point 8: Add a table or sub-section comparing acacetin with other flavonoids (e.g., apigenin, luteolin, prunin, isorhamnetin) in terms of network pharmacology targets and binding affinities.
- Response 8: We appreciate the reviewer’s suggestion. We agree that such comparisons are valuable, and thus we have incorporated textual discussion comparing acacetin with apigenin and luteolin in Section 4 (4. Network Pharmacology-based Target Prediction of Acacetin), while providing docking thresholds for pharmacological interpretation; the revised text now provides clearer context on how to interpret the pharmacological significance of docking results, thereby strengthening the comparative relevance of acacetin’s binding profiles. However, to maintain a focused scope on acacetin, we did not create a separate comparative table.
- Point 9: Improve the conclusion section by outlining a translational roadmap.
- Response 9: We thank the reviewer for this suggestion. We would like to clarify that Figure 2 already provides an integrated translational roadmap, summarizing mechanistic insights, preclinical findings, and future perspectives. To avoid redundancy, we maintained this structure, but we also clarified in the Conclusion that Figure 2 serves this role explicitly.
- Point 10: Edit the manuscript for grammar and scientific tone. Correct tense inconsistencies, remove redundancy, and refine word choice for precision. Professional language editing is required.
- Response 10: We appreciate this recommendation. We engaged the IJMS language editing service, and the full manuscript has been revised to improve grammar, tense consistency, precision and overall scientific tone.

Reviewer 2 Report
Comments and Suggestions for Authors
Here, Jang et al. present a comprehensive review on the anticancer mechanisms of acacetin, integrating network pharmacology, molecular docking, and experimental validation. Theis paper highlights acacetin’s multi-target effects on key oncogenic pathways such as EGFR, STAT3, and AKT across various cancer types. It is a very good timely review, however, below are some of my comments.
1) It would be helpful to explain why acacetin was chosen as the focal compound over other flavonoids with well-established anticancer profiles.
2) Consider adding a brief comparison of acacetin with structurally similar flavonoids (e.g., apigenin, luteolin) to highlight its unique advantages or limitations.
3) The discussion could benefit from addressing how acacetin’s multi-target activity might be influenced by tumor heterogeneity in clinical settings.
4) Adding more details on potential off-target interactions or safety concerns of acacetin would strengthen the translational value of the review.
6) The mention of doxorubicin synergy is interesting, could you expand on possible combinations with other clinically relevant therapies?
7) The reliance on in silico tools is well presented, but a critical reflection on their predictive limitations would be a valuable addition.
8) It would be useful to suggest specific biomarkers or signaling contexts where acacetin might be most effective, to guide future clinical translation.
9) Consider reproducing some figures from the previously published works.
Author Response
- Point 1: It would be helpful to explain why acacetin was chosen as the focal compound over other flavonoids with well-established anticancer profiles.
- Response 1: We thank the reviewer for this thoughtful comment. In the revised Introduction, we clarify that acacetin was selected based on a strong mechanistic foundation: direct engagement of EGFR, STAT3, and AKT; regulation beyond apoptosis (necroptosis induction, PD-L1 modulation); and anti-angiogenic effects. From a translational perspective, acacetin’s multi-target profile may help address resistance and support precision oncology. We also note that, relative to well-studied flavonoids, acacetin has received less focused attention despite emerging evidence, further motivating this review.
- Point 2: Consider adding a brief comparison of acacetin with structurally similar flavonoids (e.g., apigenin, luteolin) to highlight its unique advantages or limitations.
- Response 2: We thank the reviewer for this helpful suggestion. In the revised manuscript, we added a short comparison in Section 2.2 Pharmacological Activities of Acacetin. While apigenin and luteolin are structurally related flavonoids that also regulate major oncogenic pathways, we emphasized that acacetin shows distinctive advantages by directly targeting EGFR, STAT3, and AKT and by uniquely influencing necroptosis, PD-L1 expression, and angiogenesis. This addition highlights acacetin’s differentiated pharmacological profile compared with other closely related flavones.
- Point 3: The discussion could benefit from addressing how acacetin’s multi-target activity might be influenced by tumor heterogeneity in clinical settings.
- Response 3: We thank the reviewer for this valuable comment. In the revised manuscript, we have added a statement in Section 7. Conclusions noting that tumor heterogeneity may critically influence the clinical translation of acacetin’s multi-target activity, and highlighting the need for future studies to assess its efficacy across heterogeneous tumor contexts.
- Point 4: Adding more details on potential off-target interactions or safety concerns of acacetin would strengthen the translational value of the review.
- Response 4: We appreciate the emphasis on translation. In the revised Conclusions, we acknowledge that, although severe toxicity has not been reported in preclinical studies, off-target effects on normal cells/tissues cannot be excluded. We underscore the need for rigorous safety evaluation in future work to better establish the translational potential of acacetin.
- Point 6: The mention of doxorubicin synergy is interesting, could you expand on possible combinations with other clinically relevant therapies?
- Response 6: Thank you for the insightful suggestion. As outlined in Section 6.4, synergy with doxorubicin has been reported. We further note that inhibition of EGFR, STAT3, and PI3K/AKT, together with suppression of PD-L1 and VEGF/HIF-1α–mediated angiogenesis, provides a mechanistic rationale for combinations. Formal synergy studies remain limited; thus, combinations with widely used agents (e.g., cisplatin, paclitaxel) warrant systematic preclinical evaluation.
- Point 7: The reliance on in silico tools is well presented, but a critical reflection on their predictive limitations would be a valuable addition.
- Response 7: We appreciate this important point. At the end of Section 5.5, we now acknowledge that docking and network pharmacology are constrained by structural data quality and algorithmic assumptions and require experimental validation to confirm biological relevance.
- Point 8: It would be useful to suggest specific biomarkers or signaling contexts where acacetin might be most effective, to guide future clinical translation.
- Response 8: Thank you for underscoring biomarker-guided translation. In the revised Conclusions, we highlight biomarker-based patient stratification and provide examples of signalling contexts – such as EGFR overexpression, STAT3 hyperactivation, and AKT-dependent tumors – where acacetin may be particularly effective, while keeping the section concise.
- Point 9: Consider reproducing some figures from the previously published works.
- Response 9: We thank the reviewer for this comment and appreciate the suggestion. While we did not reproduce previously published figures to avoid redundancy, we ensured that all key mechanistic and translational information is captured in Tables 1-2 and Figure 2, which were newly created this review.
(i) Table 1 shows the mechanisms of action of acacetin across different cancer types and the up- or down-regulation of related molecules; (ii) Table 2 illustrates the predicted and validated molecular targets of acacetin based on docking and network pharmacology; and (iii) Figure 2 presents a schematic overview of its multi-target anticancer mechanisms integrating network pharmacology, molecular docking, and experimental validation.
These visualizations were specifically designed for this review to provide a comprehensive and original review.

Round 2
Reviewer 1 Report
Comments and Suggestions for Authors
The revised manuscript has been improved with the authors' updates. I recommend accepting the paper in its current form.
Reviewer 2 Report
Comments and Suggestions for Authors
The authors have addressed my queries.